ecology

foraging behaviour, sexual dimorphism, niche segregation, Westland petrel, New Zealand

**Author for correspondence:**
Timothée A. Poupart
e-mail: timothee.poupart@gmail.com

# Foraging niche overlap during chick-rearing in the sexually dimorphic Westland petrel

Timothée A. Poupart[1,2,3], Susan M. Waugh[2], Akiko Kato[3] and John P. Y. Arnould[1]

[1]School of Life and Environmental Sciences, Faculty of Science, Engineering and the Built Environment, Deakin University, 221 Burwood Highway, Burwood, Victoria 3125, Australia
[2]Museum of New Zealand, Te Papa Tongarewa, PO Box 467, Wellington 6011, New Zealand
[3]Centre d'Études Biologiques de Chizé, UMR7372 CNRS/La Rochelle Université, 79360 Villiers-en-Bois, France

 TAP, 0000-0003-4120-933X; SMW, 0000-0002-3698-588X;
AK, 0000-0002-8947-3634; JPYA, 0000-0003-1124-9330

Most Procellariform seabirds are pelagic, breed in summer when prey availability peaks, and migrate for winter. They also display a dual foraging strategy (short and long trips) and sex-specific foraging. The Westland petrel *Procellaria westlandica*, a New Zealand endemic, is one of the rare seabirds breeding in winter. Preliminary findings on this large and sexually dimorphic petrel suggest a foraging behaviour with no evidence of a dual strategy, within a narrow range and with shared areas between sexes. To investigate further this unusual strategy, the present study determined the fine-scale at-sea behaviours (global positioning system and accelerometer data loggers) and trophic niches (stable isotopes in whole blood) of chick-rearing individuals (16 males and 13 females). All individuals foraged on the shelf-slope of the west coast of New Zealand's South Island with short, unimodal trips. Both sexes foraged at similar intensity without temporal, spatial or isotopic niche segregation. These findings suggest the presence of a winter prey resource close to the colony, sufficient to satisfy the nutritional needs of breeding without increasing the foraging effort or intra-specific competition avoidance during winter. Additional data are needed to assess the consistency of foraging niche between the sexes and its reproductive outcomes in view of anticipated environmental changes.

## 1. Introduction

Seabirds are mainly represented by the Procellariiforms (36%, [1]), which are responsible for their biggest consumption of marine resources worldwide (approx. 24.1 million tons yr$^{-1}$, [2]). They range in size from the smallest storm-petrel (wing span 32 cm,

body mass 20 g) to the largest albatross (wing span >300 cm, body mass 1200 g, [3]). While their successful adaptive radiation has led Procellariiforms to occupy diverse ecological niches from tropical to polar regions [4], they still share common life-history traits linked to their pelagic lifestyle.

Within the highly spatio-temporally variable marine environment [5,6], these species rely on visual and olfactory cues [7] to forage on epipelagic fishes, squid and crustacean. To maximize prey encounters, Procellariiforms have to prospect wide areas by covering long distances within a large range [8,9]. This energetic constraint is traditionally seen as the cause of their delayed sexual maturity, low fecundity, slow growth of a single offspring and long lifespan [10].

During breeding, food intake must increase to satisfy the elevated energy expenditure of reproduction [11–13]. Breeding adults are constrained to follow a central place foraging strategy because of the need to regularly return to the nest to feed their young [14]. To maximize energy efficiency of reproduction, a dual foraging strategy is usually used during the chick-rearing period: parents alternate several short foraging trips with a long one. Short trips (1–5 d) undertaken within a small range are dedicated to maximize the provisioning of the chick, while long trips (6–29 d) reach distant more productive areas to restore parental body condition [15–17]. This bimodality in foraging trips can also buffer fluctuations in local food supply [18].

In temperate and polar environments, the alternation of seasons induces cyclic variations of ocean productivity [19], compelling most Procellariiforms to undertake migrations during the non-breeding period [20,21] to mitigate or avoid hostile conditions [22]. Correspondingly, to ensure successful chick-rearing, seabirds mostly breed in summer, when marine prey availability is usually the greatest [23,24]. Another characteristic of Procellariiforms is the similarity in plumage characteristics but differences in morphology between the sexes [25]. Male-biased sexual size dimorphism (SSD) is predominant among species with males being larger in head, bill, wing, tarsus, or tail length and body mass than females [26]. Only the storm petrels (*Hydrobatidae*) display female-biased SSD [27,28]. Size dimorphism in body mass varies from 15 to 22% in large albatrosses and giant petrels to 6–11% in smaller Procellariiform species [29]. This marked SSD induces sex-specific reproduction costs [13,30] often linked with sex-specific foraging strategies, involving spatial segregation or habitat specialization [31–33]. Consequently, males and females can potentially be sensitive to different threats, like seabird by-catch [34,35].

Few Procellariiform species (*n* = 12) differ from the general reproductive pattern, with a non-breeding period during summer, synchronous egg laying in autumn and chick-rearing throughout the winter in the Southern, the Atlantic and the Pacific oceans. As breeding phenology is a major life-history trait reflecting the species' adaptation to their environment [36], studying these unusual winter-breeding Procellariiforms may reveal an original use of oceanic habitats coupled with a peculiar winter prey. However, the remote areas occupied by these winter breeders, coupled with challenging access (winter weather, burrow nesting, nocturnal habits), has caused this group to remain the least documented [37].

One such species is the Westland petrel (*Procellaria westlandica*), an endemic of the west coast of New Zealand's South Island classified as endangered [38,39]. Preliminary studies showed that despite a wingspan of 140 cm conferring this flap-glider species with a high flight capability [40], this species has a reduced foraging range and performs shorter trips than other smaller Procellariiforms [41,42]. In addition, although males are larger than females (SSD of 6% in body mass, [43,44]), both sexes appear to forage in the same areas throughout the whole breeding cycle [45]. Furthermore, in contrast to its congenerics [46,47], Westland petrels sustain the increased nutritional requirements of chick-provisioning with no indication of bimodality in foraging trip duration [45,48].

There is currently little information on the fine-scale foraging behaviour of this species, needed to understand how its nutritional requirements for successful breeding can be obtained following a strategy vastly different to that of the majority of Procellariiforms. Such knowledge is important for understanding the potential impacts of environmental variability on this threatened species, given the anticipated shifts in the region's marine ecosystem in response to global climate change [49]. Here, we present a tracking study of Westland petrels integrating fine-scale data to test the hypothesis that this winter-breeding species has a singular foraging ecology during chick provisioning. Given the power of fine-scale movement analysis to reveal small-scale sexual segregation [50], we can expect dimorphic males and females Westland petrel to exhibit small-scale differences in their at-sea movements, activity budget, foraging behaviour or diet during winter. The objectives of the present study, therefore, were to examine in chick-rearing male and female Westland petrels: (i) fine-scale spatio-temporal foraging behaviour; (ii) the environmental factors influencing their foraging; and (iii) their associated isotopic niche.

# 2. Material and methods

## 2.1. Field procedures

Fieldwork was conducted in the coastal ranges near Punakaiki, Westland, New Zealand (42.146° S, 171.341° E). Endemic to this location, an estimated breeding population of 6200 pairs [51] nest in burrows dug in the broadleaf/podocarp rainforest soil [52]. Birds from the Scotsman's Creek study colony were captured in the nest burrow, accessible via artificial inspection hatches, over two years during the post-brood period (11–15 August 2016 and 2017). During their nocturnal attendance at the nest, one adult per burrow was captured (2016 $n = 17$, 2017 $n = 15$) after having had a chance to feed their chicks (burrows checked every hour). Individuals were weighed in a cloth bag with a suspension scale (± 25 g, Pesola) and their morphometrics taken (culmen depth, width and length, as well as tarsus length with slide callipers at the nearest 0.1 mm, and wing length with a stopped-rule ± 1 mm). Individuals were also banded (if not already) with a uniquely numbered metal band.

The bird was then equipped with a global positioning system (GPS) data logger (i-gotU GT-120, Mobile Action Technology, Taiwan, $45 \times 25 \times 12$ mm, 15 g) sampling location at 5 and 3 min intervals in 2016 and 2017, respectively, with an accuracy of ± 10 m [53] to obtain information on its at-sea movements. To obtain behaviour information, individuals were also instrumented with a tri-axial accelerometer data logger sampling at 25 Hz (Axy-Depth, Technosmart, $40 \times 15 \times 11$ mm, 7 g, also sampling pressure at 1 Hz; or X16-mini, Gulf Coast Data Concepts, $47 \times 22 \times 9$ mm, 15 g). The GPS and X16-mini data loggers were encapsulated in heat-shrink tubing (Tyco Electronics, Switzerland) and all loggers were attached to the dorsal midline feathers (accelerometer between the scapula, GPS anterior to the tail-feathers) using black waterproof tape (Tesa 4651, Beiersdorf AG). The total mass of the data loggers was 28–34 g ($2.9 \pm 0.3\%$ of body mass) such that they are likely to have had a negligible negative impact on foraging behaviour during the short-term deployment [54,55]. After instrumentation, individuals were returned to their burrow to resume normal behaviours.

Each nest was inspected the following day with a burrowscope (Sextant Technology Limited, New Zealand) through the natural entrance to confirm the instrumented bird had departed. The burrow entrance was then fitted with a one-way trap door or a wooden stick, to reveal the nocturnal visit of a parent, and checked at night every hour until recapture. Upon recapture, data loggers were removed, the birds were weighed again and a blood sample (0.3 ml) was obtained by venepuncture of a tarsal vein. The blood was immediately stored in 70% ethanol for later genetic sex determination (Parentage and Animal Genetic Services Centre, Massey University, Wellington) and stable isotope analyses (GNS Science, Lower Hutt, New Zealand). All handling of birds for procedures lasted less than 15 min. During the fledging period three months later (November) the burrows were monitored again to count the surviving chicks in order to obtain the breeding success.

## 2.2. Data processing and statistical analyses

The degree of SSD was tested on each morphometric parameter means with $t$-tests and quantified using Storer's index [56]:

$$SSD = 100 \times \frac{\text{mean males} - \text{mean females}}{(\text{mean males} + \text{mean females}) \times 0.5}.$$

This percentage of the relative difference in parameter between sexes has the advantage to avoid a size effect in the measure of SSD [57].

The GPS data were plotted in a geographical information system (ArcMap 10.2, ESRI, Redlands, CA, USA) and locations recorded on land were removed. Resulting at-sea movements were processed in the R statistical environment v 3.4.2 [58] to remove erroneous locations with a 25 m s$^{-1}$ speed filter [59]. Foraging trip metrics (duration, maximum distance from the colony, total horizontal distance travelled, average speed) were calculated using the *adehabitatHR* package [60]. Additional trip duration data were obtained during the chick-rearing period, from, respectively, 29 trips in 2012 and six trips in 2015 [45]. Trip duration distribution of these pooled trips was assessed for bimodality with calculation of the bimodality coefficient [61] and mode(s) estimation(s) by mixture distribution implemented in the *modes* package [62]. At-sea locations were linearly interpolated at 1 s intervals and the maximum convex polygon enclosing trip locations was used to estimate the home range area using the *geosphere* package [63].

At-sea behaviours (flapping flight, soaring flight, rafting on water, foraging on water and diving) were inferred at each second of the trip using the acceleration data, in order to access the individual's fine-scale activity budgets and spatio-temporal localization of foraging behaviour. Data from the seven birds equipped with Axy-Depth (tri-axial acceleration and pressure) and GPS data loggers were visualized in Igor Pro software v. 7 (Wavemetrics Inc., USA, 2000). The pressure was converted to depth ($D$ in m) using the atmospheric pressure ($P_a$ in mBar) as the baseline of measured pressure ($P_m$) in the relationship $D = 0.01(P_m − P_a)$, and corrected for surface drift. Given the high sensibility of the sensor (5 mBar) potentially causing low-pressure variance other than actual diving [64], submergences greater to the body length of the bird (0.55 m) were considered as a dive.

The dive depth and the average speed determined from GPS data were then used as references to guide the interpretation of the acceleration signals. The visual inspection of the acceleration data indicated a higher variation in the heave axis during flight (owing to wing beats) than when rafting on water [65,66]. Hence, a spectrum analysis was conducted on the heave axis data using the package *Ethographer* within Igor Pro [67], followed by a K-means clustering ($n = 2$) in order to separate periods of flapping and non-flapping, and retrieve their characteristics (dominant cycle and amplitude) with the peak tracer function. Using a combination of the dominant cycle and amplitude of flaps, with the heave value and variation associated with air and water, each second of the trip was classified as flapping flight, soaring flight or rafting on water behaviour (electronic supplementary material, figure S1). For the acceleration data in each axis, the static component of the acceleration was obtained with a running average of the raw data over 1 s [65], in order to calculate the body pitch $= \mathrm{atan}\big((Ax/\sqrt{Ay^2 + Az^2}) \times 180/\pi\big)$. This body pitch allowed us to define the foraging events when a bird was not flying and displayed a negative body pitch value below $−25°$ [68], induced by a downward head movement associated with surface-seizing, surface-diving and pursuit-plunging for prey [69]. The frequency distribution of the time interval between seconds of foraging behaviour displayed a sharp decrease for intervals greater than 5 s (electronic supplementary material, figure S2) and, therefore, this threshold was used to separate distinct foraging events.

The acceleration-based method described above identified $91.4 \pm 5.2\%$ of dives detected by pressure data, validating this method for application on the remaining individuals that were instrumented with X16-mini accelerometers. In addition, using the calculated descent rate from dive data individuals ($0.84 \pm 0.0$ m s$^{-1}$, electronic supplementary material, figure S3), the dive depth of the X16-mini individuals was estimated from the duration of the dive. Lastly, the dynamic component of the acceleration in each channel (e.g. raw acceleration minus static acceleration) was used to calculate the vectorial dynamic body acceleration (VeDBA $= \sqrt{(Ax^2 + Ay^2 + Az^2)}$), proxy of whole-body activity [70], to compare the relative movement costs of each identified.

To investigate the oceanographic variables influencing the Westland petrel habitat use, remote-sensed oceanographic variables reported to be influential for Procellariiforms [71] were gathered. Static variables such as depth and seafloor slope, and dynamic variables of the sea surface such as chlorophyll-*a* concentration, temperature and its anomaly, height, water velocity, wave height and wind were obtained at spatial resolution between 0.01° and 0.25° and temporal resolution between an hour and a day (electronic supplementary material, table S1). For each tracked individual, the time spent foraging (regrouping foraging identified on water and diving) was localized on the interpolated track and converted into a standardized percentage within square grid cells of 0.04° (*ca* 4 km$^2$), matching the most accurate spatio-temporal combination available for oceanographic variables in order to approach real-time at-sea conditions. The resulting standardized time spent foraging in area was then spatio-temporally matched with the oceanographic variables using the R package *raster* [63].

Stable isotope analyses of nitrogen ($\delta^{15}$N) and carbon ($\delta^{13}$C), proxies of trophic ecology [72], were used to infer the trophic level of individuals and quantify their niche width [73]. Analysis was conducted on whole blood, a tissue integrating the turnover rates of both plasma and red blood cells, representing the assimilated diet up to four to five weeks prior to sampling in Procellariiforms [74]. Samples were oven-dried for 24 h at 60°C, grounded and homogenized, before approximately 0.5 mg was packed into tin capsules for combustion in a Eurovector elemental analyser coupled to an Isoprime mass spectrometer (GV Instruments, UK). Their isotopic deviation was defined as $\delta(‰) = [(R_s − R_{ref})/R_{ref}]$, where $R_s$ is the isotopic ratio measured for the sample and $R_{ref}$ is the reference standard. Internal laboratory standards (leucine; $\delta^{13}$C $−28.30‰$ and $\delta^{15}$N $+ 6.54‰$; ethylenediaminetetraacetic acid (EDTA): $\delta^{13}$C $−31.12‰$ and $\delta^{15}$N $+ 0.58‰$; caffeine, $\delta^{13}$C $−38.17$ $1‰$ and $\delta^{15}$N $−7.82‰$; cane sugar $\delta^{13}$C $−10.33‰$) calibrated to primary reference materials (RM's, International Atomic Energy Agency (IAEA)-N1, $\delta^{15}$N $+ 0.43‰$; IAEA-N2, $\delta^{15}$N $+ 20.41‰$; IAEA-CH6, $\delta^{13}$C $−10.449‰$; and IAEA-CH7, $\delta^{13}$C $−32.151‰$) relative to international standards $\delta^{13}$C$_{VPDB}$ and $\delta^{15}$N$_{Air}$ indicated an analytical precision of $\pm 0.1‰$ for $\delta^{13}$C and $\pm 0.2‰$ for $\delta^{15}$N.

**Table 1.** Logger deployments on Westland petrels and their GPS trip metrics during post-guard stage (mean ± s.e.).

| | 2016 | | 2017 | |
|---|---|---|---|---|
| | males | females | males | females |
| individuals equipped (*n*) | 11 | 6 | 7 | 8 |
| individuals (trips) with GPS and acceleration | 9 (9) | 5 (5) | 7 (9) | 8 (10) |
| body mass (kg) | 1.25 ± 0.03 | 1.21 ± 0.02 | 1.27 ± 0.05 | 1.25 ± 0.04 |
| trip duration (h) | 46 ± 6 | 55 ± 16 | 41 ± 6 | 60 ± 9 |
| max. distance from the colony (km) | 120 ± 12 | 106 ± 33 | 173 ± 28 | 200 ± 37 |
| total horizontal distance covered (km) | 749 ± 98 | 677 ± 234 | 890 ± 155 | 1276 ± 247 |
| mean horizontal speed (km h$^{-1}$) | 16.8 ± 1.0 | 10.8 ± 1.6 | 21.7 ± 1.3 | 19.8 ± 1.2 |
| mean VeDBA (g) | 0.221 ± 0.007 | 0.216 ± 0.006 | 0.239 ± 0.012 | 0.219 ± 0.006 |

All statistical analysis were performed within the R statistical environment v. 3.4.2 [58]. The effect of mass change on the foraging rate (number of foraging events divided by the trip duration), and the effect of sex and year on body mass, mass change and stable isotope values, were investigated with analysis of variance (ANOVAs) validated by the examination of their residuals [75]. As the GPS trip metric data included multiple trips by some individuals, the influence of sex and year was investigated with linear mixed models (LMMs). They were performed with individual identity as a random factor to account for their pseudo replication [76] using the *nlme* package [77]. Similar LMMs were also conducted on the activity budget (e.g. proportion of time flapping, gliding, rafting on water, foraging). The temporal variation of the activity budget and dive depth throughout the day was further investigated with generalized additive mixed models (GAMMs) using the *mgcv* package [78]. In addition, GAMMs were used to investigate the influence of the encountered oceanographic parameters on the log-transformed time spent foraging by each sex. Prior to inclusion in the model, oceanographic parameters were scaled and checked for collinearity, with a cut-off criterion of $|r_s|$ = 0.5 for inclusion in the model.

Model selection was conducted by comparison of the Akaike information criteria (AIC, [79]) between candidates models using the *MuMin* package [80]. The best supported model (ΔAIC = 0) without other equally supported models having ΔAIC < 4, [81], or by default, the most parsimonious model (constituted by the important explanatory variables identified by model averaging procedure among the equally supported models, [81]) was retained (electronic supplementary material, table S2). The validation of the retained models was based on the examination of their residuals [75]. The isotopic niche quantification was performed using the ellipse-based metrics of the *SIBER* package [82], by the standard ellipse area corrected for sample size (SEA$_C$) and Bayesian standard ellipses area (SEA$_B$). Unless otherwise stated, all data are presented as mean ± s.e.

# 3. Results

## 3.1. At-sea movements and activity budgets

At-sea movement (GPS) data were obtained from 32 individuals, and owing to device malfunctions, at-sea movements combined with behaviour data were obtained from 29 individuals (16 males/18 trips, 13 females/15 trips; table 1) totalling 1556 h in duration. Morphometric measurements confirmed a male-biased structural dimorphism (table 2). However, the body mass at instrumentation (1250 ± 19 g) did not differ significantly between sexes and years (ANOVA, $p > 0.5$ in both cases). At recapture, some individuals gained mass (+94 ± 15 g, $n = 14$) and some lost mass (−113 ± 13 g, $n = 16$). Mass change was not significantly different between sexes and years (ANOVA, $p > 0.2$ in both cases).

Individuals foraged up to 154 ± 14 km away from the colony over the shelf-slope and submarine canyons off the west coast of New Zealand's South Island (figure 1). Their trips lasted 50 ± 4 h, covered a total horizontal distance of 920 ± 95 km at an average speed of 18 ± 1 km h$^{-1}$ that resulted in a home range size of 36 563 km$^2$ in 2016 and 80 052 km$^2$ in 2017. The distribution of foraging trip durations did not reveal any indication of bimodality with a bimodality coefficient of 0.38 (suggesting

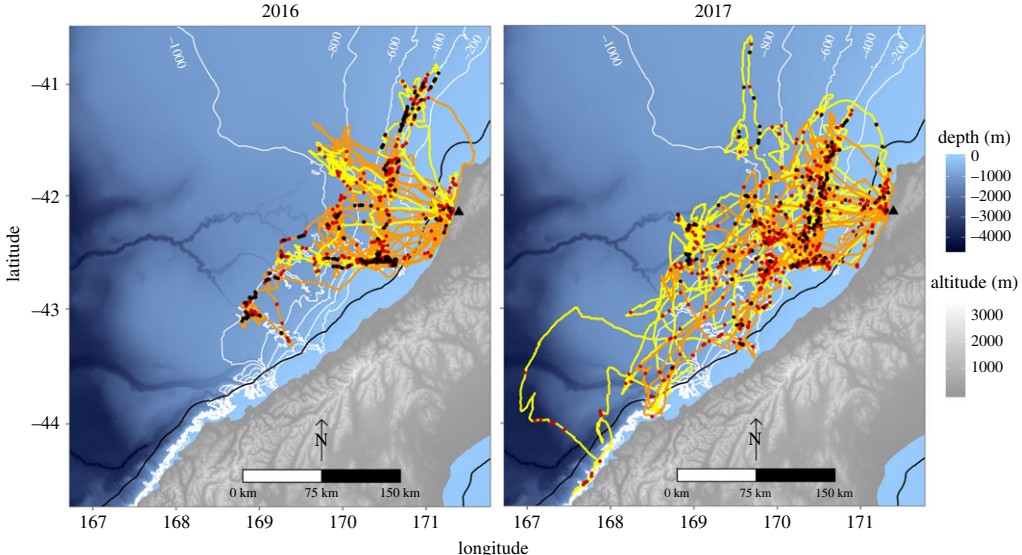

**Figure 1.** At-sea movements and foraging behaviour of breeding Westland petrels from Punakaiki (New Zealand) during the post-guard stage. Tracks are shown in orange for males and yellow for females. On top of the tracks, the red dots represent the surface foraging and the black dots the dives. On land, the black triangle represents the colony location, and at sea, the black line is the 12 nautical miles boundary of the Territorial Sea.

**Table 2.** SSD in morphometric measurements of genetically sexed Westland petrels tracked during rearing chicks in 2016 and 2017.

| | males ♂ | females ♀ | *t* test result | | |
|---|---|---|---|---|---|
| | (*n* = 18) | (*n* = 14) | *t* value | (*p*) | Storer' SSD index (%) |
| culmen depth (mm) | 18.2 ± 0.3 | 16.4 ± 0.2 | −4.8 | <0.001 | 10.4 |
| culmen width (mm) | 23.6 ± 0.2 | 22.2 ± 0.3 | −2.1 | <0.01 | 6.1 |
| culmen length (mm) | 51.1 ± 0.4 | 48.8 ± 0.4 | −3.8 | <0.001 | 4.6 |
| wing length (mm) | 392 ± 3 | 379 ± 2 | −3.5 | <0.01 | 3.5 |
| tarsus length (mm) | 65.2 ± 0.5 | 64.0 ± 0.6 | −1.4 | 0.1 | 1.8 |
| mass at deployment (g) | 1256 ± 25 | 1238 ± 28 | −0.4 | 0.6 | 1.4 |

unimodality) and a single mode at 3 days (figure 2). The LMMs revealed no significant influence of year and sex on the total horizontal distance travelled, trip duration or mean VeDBA. Year of study had a significant influence on the maximum distance from the colony and the average speed, with trips reaching maximum distance greater by 69 ± 28 km ($p = 0.02$), travelled at speeds 6.7 ± 1.3 km h$^{-1}$ faster in 2017 ($p < 0.001$). Individual sex had a significant influence only for the average speed, with males travelling faster by 3.6 ± 1.3 km h$^{-1}$ than females ($p = 0.01$, table 1).

Individual activity budgets comprised 51.4 ± 2.5% flapping flight, 34.6 ± 2.7% rafting on water, 13.7 ± 1.0% soaring flight and only 0.13 ± 0.01% of time foraging. These behaviours had different VeDBA values, the highest being for foraging (0.73 ± 0.06 g), followed in decreasing order by flapping flight (0.39 ± 0.01 g), gliding flight (0.20 ± 0.01 g) and rafting on water (0.19 ± 0.01 g). The GAMMs revealed that these behaviour proportions all varied significantly according to the time of day ($p < 0.001$). Rafting on water peaked at the middle of the day, at the expense of flapping and gliding flight that both increased at dawn and dusk (figure 3). The sex did not significantly influence the timing of these activities at sea, and significant inter-annual variation occurred only with rafting on water and flapping flight behaviours, which decreased by 13% and increased by 11%, respectively, in 2017 (table 3).

## 3.2. Fine-scale foraging behaviour and isotopic niche

The foraging behaviour, inferred during a total of 7348 s (2 h), occurred during 4080 foraging events (between 8 and 399 events per trip, according to their duration). The resulting foraging rates were

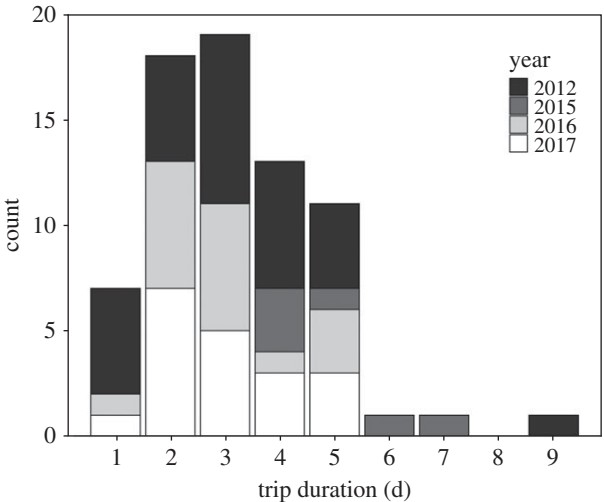

**Figure 2.** Distribution of the foraging trip durations of chick-rearing Westland petrels. Data from 2016 and 2017 were gathered during the present study, data from 2012 and 2015 are from Waugh *et al.* [45].

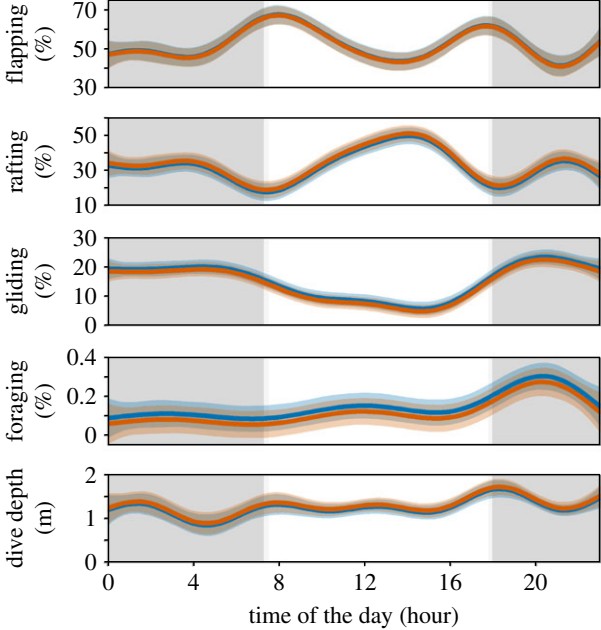

**Figure 3.** Temporal variation of the at-sea time activity budget of Westland petrels. The proportions of behaviours and dive depth are predicted on an hourly basis by generalized additive mixed models (GAMMs) for males (blue lines) and females (orange lines). The shaded areas along the curves represent the associated 95% confidence interval. In the background, the dark grey and light grey shaded areas correspond to the night-time and sunset/sunrise times, respectively.

$2.7 \pm 0.2$ events $h^{-1}$ (range 0.6–5.9), equating to a foraging event every 6 km on average (range 2.5–20). These rates were not significantly influenced by sexes, years, trip duration/travelled distance (null models selected, electronic supplementary material, table S2) or by the mass change of individuals (ANOVA, $p = 0.3$). The time spent foraging consisted of $66.6 \pm 3.1\%$ surface foraging and $33.4 \pm 3.1\%$ by diving. Surface foraging events were of brief duration ($1.2 \pm 0.01$ s) and dives ($n = 152$ recorded, $n = 827$ estimated) were characterized by duration of $3.1 \pm 0.1$ s and depth of $1.3 \pm 0.0$ m. Maximum dive duration and depth reached 26 s and 8.6 m, respectively. The GAMM modelling revealed that the proportion of time spent foraging peaked during the first part of the night ($p < 0.001$; figure 3). The dive depth was not significantly influenced by the time of the day and the sex (figure 3) but by year, with depths shallower by $0.4 \pm 0.1$ m in 2017 ($p = 0.01$).

Individuals spent 75.4% of their foraging time over the peri-insular shelf-slope (200–1000 m), and only 13.3% in bathyal waters (1000–2000 m), 11.0% in neritic waters (0–200 m) and 0.09% in abyssal waters (greater than 2000 m). Foraging locations were situated above a sea floor slope

**Table 3.** Summary of the generalized additive mixed models (GAMMs) outputs used to assess the trends in the at-sea behaviour of Westland petrels provisioning chicks at Punakaiki, New Zealand. (Significant *p*-values are highlighted in italics. Edf, estimated degrees of freedom.)

| response variable | predictor variable | parametric coefficients | | | significance of smooth terms | | |
|---|---|---|---|---|---|---|---|
| | | estimate | s.e. | *t* | Edf | *F* | *p* |
| % foraging | intercept | 0.13 | 0.01 | 7.0 | — | — | *<0.001* |
| | hour | — | — | — | 1.0 | 14.8 | *<0.001* |
| % rafting on water | intercept | 43.3 | 4.5 | 9.6 | — | — | *<0.001* |
| | hour | — | — | — | 8.7 | 15.2 | *<0.001* |
| | year 2017 | −13.4 | 4.9 | −2.7 | — | — | *0.006* |
| | sex male | −6.0 | 4.9 | −1.2 | — | — | 0.2 |
| % flapping flight | intercept | 43.7 | 4.8 | 9.0 | | | *<0.001* |
| | hour | — | — | — | 8.6 | 12.6 | *<0.001* |
| | year 2017 | 11.1 | 5.3 | 2.0 | | | *0.03* |
| | sex male | 4.3 | 5.3 | 0.8 | | | 0.4 |
| % gliding flight | intercept | 14.7 | 1.0 | 14.0 | — | — | *<0.001* |
| | hour | — | — | — | 7.7 | 25.4 | *<0.001* |
| dive depth | intercept | 1.4 | 0.1 | 14.0 | — | — | *<0.001* |
| | year 2017 | −0.4 | 0.1 | −2.5 | — | — | *0.01* |

**Table 4.** Summary of the generalized additive mixed models (GAMMs) outputs used to assess the oceanographic variables influencing the foraging of Westland petrels provisioning chicks at Punakaiki, New Zealand. (Significant *p*-value are highlighted in italics. SST, sea surface temperature; ASST, sea surface temperature anomaly.)

| response variable | predictor variable | parametric coefficients | | | significance of smooth terms | | |
|---|---|---|---|---|---|---|---|
| | | estimate | s.e. | *t* | Edf | *F* | *p* |
| log(time spent foraging) females | intercept | 0.32 | 0.7 | 0.4 | | | 0.6 |
| | slope | — | — | — | 7.5 | 2.0 | *0.02* |
| | SST | — | — | — | 3.8 | 2.4 | *0.05* |
| | depth | — | — | — | 1.0 | 0.8 | *0.04* |
| | ASST | — | — | — | 2.0 | 0.8 | 0.3 |
| log(time spent foraging) males | intercept | 0.5 | 0.1 | 2.6 | — | — | *0.007* |
| | SST | — | — | — | 2.4 | 0.7 | 0.5 |
| | ASST | — | — | — | 2.9 | 1.0 | 0.3 |
| | wind | — | — | — | 1.0 | 3.0 | 0.08 |
| | wave | — | — | — | 5.0 | 1.7 | 0.1 |
| | current | — | — | — | 1.0 | 5.7 | *0.01* |
| | depth | — | — | — | 1.0 | 3.8 | *0.05* |
| | slope | — | — | — | 2.0 | 2.0 | 0.1 |

ranging from 0.02 to 15.00° and where the chlorophyll-*a* concentration was $0.35 \pm 0.01 \, \mathrm{mg \, m^{-3}}$. The sea surface was characterized by a temperature ranging from 10.6 to 14.1°C with associated temperature anomalies between −1.8 and 0.9°C. The sea surface height varied between 0.03 and 0.18 m and was agitated by currents up to $0.3 \, \mathrm{m \, s^{-1}}$, waves ranging from 0.6 to 4.8 m, and wind blowing between 2 and $14 \, \mathrm{m \, s^{-1}}$. The GAMMs revealed a significant influence of four oceanographic variables on the time spent foraging, with results differing between sexes (table 4 and figure 4).

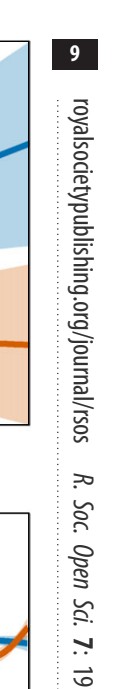

**Figure 4.** Relationship between the time spent foraging in area by Westland petrels provisioning chicks and influencing oceanographic parameters. Lines represent the generalized additive mixed models (GAMM) for males (in blue) and for females (orange). The shaded areas show the associated 95% confidence intervals.

**Table 5.** Isotopic values (whole blood, mean ± s.d.) of male and female Westland petrels provisioning chicks in 2016 and 2017.

| year | sex | $n$ | $\delta^{13}C$ (‰) | $\delta^{15}N$ (‰) |
|---|---|---|---|---|
| all individuals | | 34 | −18.3 ± 0.2 | 15.4 ± 0.2 |
| 2016 | male | 12 | −18.2 ± 0.3 | 15.5 ± 0.2 |
| 2016 | female | 7 | −18.3 ± 0.3 | 15.4 ± 0.2 |
| 2017 | female | 8 | −18.3 ± 0.2 | 15.4 ± 0.2 |
| 2017 | male | 7 | −18.4 ± 0.4 | 15.5 ± 0.2 |

For females, the time spent foraging was negatively influenced by positive temperature anomalies ($p = 0.001$) and negatively by sea surface temperature ($p < 0.05$). For males, it was positively influenced by wind speed ($p < 0.01$) and positively influenced by current speed ($p = 0.01$).

Stable isotope values of whole blood (mean ± s.d.) were 15.4 ± 0.2‰ for $\delta^{15}N$ and at −18.3 ± 0.3‰ for $\delta^{13}C$ (table 5). These isotopic results showed little variation and did not significantly change between years or sexes (ANOVA, $p \geq 0.1$ in all cases). All values, pooled together to construct the isotopic niche, revealed a small isotopic niche (figure 5) with an area estimated at 0.23‰$^2$ by the SEA$_C$ and at 0.21‰$^2$ by the SEA$_B$. These feeding conditions related to low fledging success in 2016 (51%) and high fledging success in 2017 (85%), as indicated by the burrow monitoring at the colony with an average success of 62% [83].

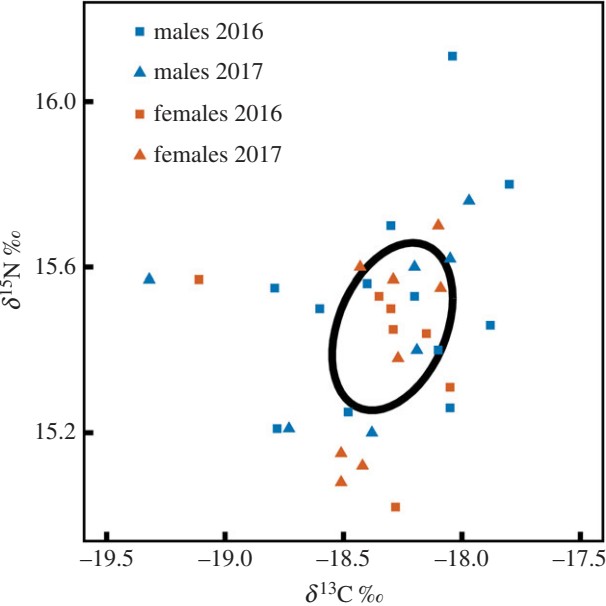

**Figure 5.** Isotopic values and niche of breeding Westland petrels. The colours and point shapes indicate the sex and year respectively. The isotopic niche is represented by the 40% corrected standard ellipse (SEA$_C$).

## 4. Discussion

Using combined GPS and accelerometer data from adult males and females rearing chicks during two breeding seasons, the present study provided information on the fine-scale foraging behaviour of Westland petrels. The results confirmed the species' small foraging area over the shelf-slope of New Zealand's South Island west coast, exploited by relatively short foraging trips unimodal in duration. The acceleration data showed that foraging behaviour comprised short and dispersed events, representing a small proportion of their at-sea activity budget. The analysis of movements and isotopic niche did not reveal clear sexual segregation despite their SSD. However, sexual differences were observed in the influence of oceanographic variables on time spent foraging, suggesting a potential small-scale habitat specialization. These findings suggest male and female Westland petrels consume the same prey, sufficiently abundant locally during winter to satisfy the nutritional needs of breeding without intra-specific competition avoidance or increased foraging effort compared to summer breeding species.

### 4.1. At-sea movements and activity budget

The present study confirmed and refined the chick-rearing movements of Westland petrels previously reported by at-sea observations [84] and tracking studies [45,48]. The species displayed restricted breeding movements ($432 \pm 20 \, \text{km} \, \text{d}^{-1}$) during the present study compared to the non-breeding movements ($1351 \, \text{km} \, \text{d}^{-1}$, [44]). This is consistent with the need for regular returns to the nest for feeding the chick [14]. However, the small maximum distance to the colony and home range area values recorded in the present study are in contrast with other Procellariiforms during breeding. For instance, the similar-sized congeneric white-chinned petrel *Procellaria aequinoctialis* displays much wider movements and exploits an area eight times bigger [46,85,86]. Even smaller Procellariidae and Hydrobatidae, species with lower flight capacity, display greater range and exploit a wider area [59,87,88] than the Westland petrel.

In addition to this particular foraging distribution, the Westland petrel at-sea movements were also different to many Procellariiforms as a result of the absence of bimodality in their duration [16]. The alternating of short trips close to the colony with long trips to distant more productive areas, respectively, associated with parental mass loss and gain [15,89] is reported to be advantageous to reduce parent–offspring energy conflict [90]. Both unimodal and dual strategies are reported in shearwaters with unimodality exhibited in high-quality habitats and bimodality when parents face low prey availability close to the colony [91–93]. Despite the presence of reachable areas of enhanced

productivity favourable for Procellariiforms near New Zealand, such as the subtropical front [87,94] and the polar frontal zone [95], individuals in the present study exploited only a small foraging range. This suggests the shelf-slope of New Zealand's South Island west coast provides the closest reliable prey-rich habitat for the species [96]. Nonetheless, both positive and negative mass changes were recorded during foraging trips of the present study. Although these mass changes could be biased by different recapture (and weighing) times after the individuals' return, they show greater variation than the reported average meal mass (37 g, [97]). Hence, such mass changes may result from an alternation between chick-provisioning and self-maintenance purposes during unimodal trips within the same area. A more accurate weighing of the individuals, just after their return at the colony, is required to verify this potential explanation for such contrasted mass changes.

Information on the activity budgets of foraging seabirds provides useful information on their energy expenditure, foraging effort, reproductive costs, prey availability and habitat quality [98–101]. The tracked individuals in the present study spent the majority of their time at sea in flight, and especially flapping flight, which is consistent with the reported flap-gliding flight of *Procellaria* [40]. Despite the proximity of the foraging areas, individuals in the present study spent a majority of time in flight searching for prey and little time actually foraging. This is a common pattern for albatrosses [102], petrels [87] and shearwaters [95,103] that reveal a strategy to maximize prey encounter when foraging on patchy prey [8].

The activity budgets of individuals in the present study were primarily influenced by the time of the day, and by the year of study. With time spent foraging slightly increasing during the middle of the day and peaking in the first part of the night, these findings are consistent with the constraint of available light on these visual foragers influencing their at-sea activities during the day [104]. The inter-annual variations observed in both at-sea movements and activity budget coincided with different environmental conditions. The difference in sea surface temperature ($12.8 \pm 0.03$°C in 2016, $13.4 \pm 0.01$°C in 2017) is likely to have changed the prey distribution on the New Zealand shelf [105]. The difference in wind speed ($5.5 \pm 0.08$ m s$^{-1}$ in 2016, $8.1 \pm 0.08$ m s$^{-1}$ in 2017) could explain the further maximum distance, higher trip speed and more time spent flying in 2017. Hence, the absence of difference in total horizontal distance travelled between years seems to indicate an easier prey accessibility in 2017, and suggests some behavioural plasticity to compensate for variation of the energetic costs of movements for foraging [9,106,107].

## 4.2. Foraging behaviour and trophic niche

The acceleration data gathered in the present study enabled the foraging behaviour of individuals to be accurately determined during the foraging trips. The present study revealed that the Westland petrel was able to feed at any time, while foraging is generally reduced at night for albatrosses and some petrels [95,104]. The peak observed during the first hours of darkness suggests a pulsed food availability at the beginning of the night, potentially owing to the diel vertically migrating prey reaching near surface waters [108]. Indeed, previous studies on Westland petrel diet reported feeding upon vertically migrating and bioluminescent species of fish and squid [97,109] able to be detected from long range (50–100 m, [110]) and representing important biomass on shelf-slope habitat [111,112].

Independently of the characteristics of their trips, all individuals spent only a small proportion of their time foraging. This foraging time, spread out along the tracks, was separated by intervals skewed towards short time. Therefore, these findings suggest Westland petrels feed on patchily and aggregated prey distributed throughout the shelf-slope, in accordance with what is known about the prey species of its diet [97,113]. This foraging behaviour was found to be constant in all trips for both sexes, as shown by the mean VeDBA and foraging rate at the trip scale. This is in contrast with the distinction between fine-scale and coarse-scale foraging trips reported in Scopoli's shearwater *Calonectris diomedea*, the former being more undertaken by males and having more intensive search behaviour, based on GPS clustering analysis [114]. The foraging rate observed in the present study was higher than that reported in the Scopoli's shearwater in the Mediterranean Sea (1.7 events h$^{-1}$, [64]) but similar to the southern Buller's albatross (*Thalassarche b. bulleri*, 2.5 events h$^{-1}$) which also forages in winter along the west coast of New Zealand South Island [115].

The maximum dive depths recorded in the present study were very close to that previously recorded with capillary-tube maximum depth gauges [41]. The mean and maximum dive depths recorded in the present study were shallower than those reported for congenerics, petrels and shearwaters also breeding in the New Zealand region [87,95,116] as well as worldwide (table 4 in [117]). Assuming that the Westland petrel has comparable diving capacity to the similar-sized

white-chinned petrel, the shallow dives recorded in the present study suggest an availability of different prey near the surface throughout the day.

The foraging activity observed in the present study was concentrated on the shelf-slope near the breeding colony, which receives an eastward flow from the Tasman Sea and undergoes a seasonal winter mixing and intermittent wind-driven upwelling that sustains the local food web during winter to create a prey-rich ecosystem [118,119]. Use of similar habitats was reported for the Westland petrel during its summer non-breeding period in the Humboldt Current system off South America, where at-sea observations were concentrated in waters of 200–1000 m depth undergoing upwelling [120]. The investigation of the time spent foraging by individuals in the present study in relation to oceanographic variables revealed a significant influence of depth, slope, sea surface temperature and current. Slope was also reported to increase the foraging intensity estimated by kernel density estimate [45]. These findings are consistent with the slope and current involved in horizontal advection, uplifting and mixing of subsurface waters in the area [121].

The $\delta^{13}C$ values observed in the present study are consistent with feeding in subtropical waters [122]. The $\delta^{15}N$ values confirmed that the Westland petrel was feeding at high trophic level [123] in accordance with the previously described fish-dominated diet of the species [97] and congenerics [124,125]. Chick-rearing individuals showed comparable isotopic values in their whole blood to incubating birds and a comparable isotopic niche area, with the exception of the larger $SEA_C$ observed in 2015 during a strong El Niño year [45]. This is in contrast with smaller Procellariiforms that have been found with altered $\delta^{15}N$ over the breeding season [126]. Hence, the Westland petrel seems to be relying on a narrow trophic resource, which is persistent throughout its breeding cycle during winter.

## 4.3. Lack of sexual segregation

Despite the male-biased SSD present in the Westland petrel, the only difference in foraging trip parameters found in the present study between the sexes was the higher average flight speed of males. This result is in accordance with their longer wings ([43], table 2) and the close link between wing morphology and flight behaviour [127]. Males and females also used similar foraging areas over the shelf-slope, with both sexes foraging at same rate and during the same time of day. Their resulting isotopic niches also indicated an absence of sex-specific dietary specialization. Together, these results suggest a similar foraging strategy for both sexes, which is consistent with previously reported lack of difference for the species in foraging area estimates [45] and their similar survivorship [128]. These findings contrast with observations for many Procellariiforms (and other seabirds) which display sex-specific spatial segregation and foraging specialization reducing intersexual competition [129–132].

The only sexual difference associated with foraging in the present study was related to habitat use, with differences in the influence of current, slope and sea surface temperature on the time spent foraging between the sexes. These findings suggest slight sexual differences in fine-scale habitat use, as has been reported during the winter chick-rearing in the wandering albatross *Diomedea exulans* [50], which could relate to their morphologic dimorphism. A lack of spatial segregation as observed in Westland petrels during the pre-laying, incubation and chick-rearing periods [45], has also been reported in Cory's shearwater *Calonectris borealis* off the west coast of Africa [91] within a major upwelling zone [133]. Under conditions of high habitat quality and prey abundance, both sexes can forage on the same niche without competition, and SSD becomes supported mainly by sexual selection [91,134]. The investigation of the other hypothesis for SSD (e.g. the niche divergence hypothesis [135,136]) would require further studies on colony attendance, incubation behaviour and chick provisioning by both sexes in Westland petrels to assess their respective reproductive roles [137,138].

## 4.4. Conservation

The reliance on the relatively small foraging area identified in the present study could pose problems for the species. The Westland petrel population size is small (*ca* 6200 breeding pairs [51]), and, while breeding pairs have a relatively a high fledging success, its breeding frequency is variable with some adults skipping years between attempts [83]. In addition, the long fledging period for the species, similar to other winter-breeding Procellariiforms [37], may reflect the small meal sizes delivered by breeding parents [97]. A single foraging area is likely to increase the sensitivity of a specialized predator to local food shortage [139,140]. In addition, the observed absence of sex-specific foraging strategy may limit ability of the species to adapt to expected large-scale environmental changes [49].

Additional studies are needed to evaluate the consistency of foraging behaviour by both sexes throughout the annual cycle and under variable environmental conditions (e.g. strong El Niño/La Niña years). Such data would provide insights into potential energy constraints for the species [141] and how it may respond to anticipated environmental changes [142].

The foraging area identified in the present study supports the defining of the pelagic important bird area (IBA) and marine protected area (MPA) proposed for the Westland petrel [143], where the Hoki *Macruronus novaezelandiae* trawling fishery area coincides with the species foraging areas. Despite an opportunistic usage of fisheries waste by the Westland petrel [45,97,144], fine-scale data on the movements of fishing vessels would allow future study to refine the habitat modelling with natural foraging behaviour data only. In addition, the identification of birds' interactions with fishing vessels is required to better understand their use of this alternative food resource, which may mitigate the variation of their natural prey. Such knowledge would be beneficial for the species, classified 'at risk' to by-catch [145], and which faces other fishing fleets and catch methods during their migration and non-breeding period in South American waters [44,146].

In summary, the results of the present study showed that the nutritional needs of breeding can be satisfied during winter within a restricted foraging range without a bimodal foraging trip duration or higher foraging effort than other Procellariiforms. These findings support the hypothesis of a singular foraging ecology of winter-breeding species, relying on a predictable prey field available locally. Fine-scale data allowed us to invalidate the prediction of sex-specific foraging strategy, that was expected from intra-specific competition. The large ecological niche overlap that can happen between dimorphic males and females during chick provisioning suggests sex-specific foraging as a consequence of prey accessibility. Therefore, investigating sex-specific strategy among species and populations could inform on their degree of adaptation to the prey field and their capacity to buffer large-scale environmental changes.

Ethics. All work was conducted under permit WC-26677-FAU issued by the New Zealand Department of Conservation, with approved animal handling procedures.

Data accessibility. Data available from the Dryad Digital Repository: https://dx.doi.org/10.5061/dryad.w9ghx3fmf; and https://dx.doi.org/10.5061/dryad.xpnvx0kcd [147].

Authors' contributions. S.M.W., J.P.A. and T.P. conceived the study design. T.P. collected data and wrote the paper. J.P.A., S.M.W. and A.K. substantially edited the paper. T.P., J.P.A. and A.K. developed methods. T.P. and A.K. analysed the data. S.M.W. and J.P.A. contributed substantial materials, resources and funding.

Competing interests. We have no competing interests.

Funding. The study was funded by the Museum of New Zealand Te Papa Tongarewa, Deakin University, Centre d'Études Biologiques de Chizé /University of La Rochelle and the Brian Mason Trust.

Acknowledgements. We thank Ngāti Tahu's Kati Waewae runaka for consent to study the petrels, the DOC and Conservation Volunteers New Zealand for the access in specially protected area of the Paparoa ranges. We are also grateful to all fieldworkers for their assistance with the colony monitoring and loggers' deployments (Leon Berard, Laureline Durand, Francois Même, Kerry-Jayne Wilson, Leon Dalziel), and Matt Pinkerton for his advice. We thank the associate editor Dr Agustina Gómez-Laich and two anonymous reviewers who helped to improve the manuscript.

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
