## [Reviewer comments · Royal Society Open Science]

Review History

RSOS-191511.R0 (Original submission)

Review form: Reviewer 1

Is the manuscript scientifically sound in its present form?

Yes

Are the interpretations and conclusions justified by the results?

Yes

Is the language acceptable?

Yes

Do you have any ethical concerns with this paper?

No

Have you any concerns about statistical analyses in this paper?

No

Recommendation?

Accept with minor revision (please list in comments)

Comments to the Author(s)

Globally the manuscript is well written, with a good English, which allows a fluid and comprehensive reading. All sections have enough detail to be fully understood.

Major comments:

At the end of the introduction, after describing the main objectives of their study, the authors should raise few study hypothesis followed by the corresponding expected results. It's true the existing knowledge on the foraging ecology of their study species is scarce, but there is quite a few number of studies focusing on other (similar) gadfly petrels in other study areas. Then, latter in the results and discussion, the reader will be able to check whether the results confirmed the expectations, or the authors advance and discuss an alternative reasoning behind the foraging ecology choices of Westland petrels.

P5L48-56: I realise the authors describe in detail at the supplementary Table S1 the origin of the oceanographic variables used on their work, though they should also briefly describe here the spatio-temporal resolution of those environmental predictors.

Minor corrections:

P2L54: Please replace by "Procellariiforms"
Change accordingly elsewhere in the main body of the manuscript.

P3L2: "Procellariiforms"

P3L27: "...during the chick-rearing period..."

P3L33: Replace "mediate" by "buffer"

P3L46: "...different threats, like seabird by-catch (Gianuca...)"

P3L47: Remove "A" at the beginning of the sentence

P5L2: Replace "The distribution of the trip duration distribution was assessed for bimodality..." by "Trip duration distribution was assessed for bimodality..."

P6L26: Rewrite as "...by each sex."

P7L45 and L46: remove "in area", it's kind of redundant

P7L48: "...current strength."

P10L10: Add a space before "Hence..."

P10L40: "...off the West coast of Africa..."

P10L60: "...strong El Niño/ La Niña..."

P17 Table 2: Please add an extra column with the t-test values before the column with P-values.

P24 Figure 4: Add to the legend what the two lines and CI are representing. Males and females? Change the colours; these are too similar to be easily distinguishable.

Review form: Reviewer 2

Is the manuscript scientifically sound in its present form?

Yes

Are the interpretations and conclusions justified by the results?

Yes

Is the language acceptable?

Yes

Do you have any ethical concerns with this paper?

No

Have you any concerns about statistical analyses in this paper?

No

Recommendation?

Accept with minor revision (please list in comments)

Comments to the Author(s)

RSOS-191511

"Foraging niche overlap during chick-rearing in the sexually dimorphic Westland petrel"
by Poupart T.A. et al.

This study examined sexual differences in foraging behavior of Westland petrels in details. Somewhat, the scope is quite narrow, likely a description of specific behavior in a single species and not solely contributed to understanding of the general ecology of Procellariiform seabirds. However, otherwise, the manuscript is overall well and carefully written referring to previous knowledge efficiently, and methods and analysis are comprehensive and adequate. Furthermore, discussion on the lack of sexual differences in behavior in terms of habitat quality is relatively speculative based on Louzao et al. (2011), but still understandable. So, I highly appreciate this study and believe that this manuscript can be suitable for publication in Royal Society Open Science, but it requires some reconsideration.

Specific comment

INTRODUCTION

P2 "A few Procellariiform species (n=12) differ...throughout the winter."

>The general ecology of seabirds/Procellariiforms is summarized before this sentence. These descriptions are true, but only related to a part of this study, such as size dimorphism and dual foraging strategy. For example, results of this study do not provide an insight into why Westland petrels breed in winter. So, I suggest to shorten these sentences. Especially, although I like it, the first sentence describing the diversity of seabirds does not seem necessary (too general and not related to results of this study).

DISCUSSION

P7 "without...increased foraging effort compared to summer breeding species."

>Body size, prey species, and foraging range etc. are different to species mentioned (Scopoli's shearwater and Buller's albatross). So, foraging effort cannot be simply comparable to that of Westland petrels.

P7 "both positive and negative mass changes...self-maintenance in the same area."

>Were there any differences in foraging behavior between trips with positive and negative mass changes?

P7 "Information on the activity...their energy expenditure,"

>I wonder why the authors examined VeDBA in this study. Probably, you can find results in previous studies, showing differences in energy expenditure among behaviors in congeneric or similar species (foraging, flapping, gliding, rafting). I would like to see some discussion for VeDBA, recorded for Westland petrels in concert with GPS data. Also, did overall VeDBA differ between years with different sea surface temperature conditions? Such information could be useful to predict the effect of environmental changes on their breeding success.

Decision letter (RSOS-191511.R0)

Dear Mr Poupart,

The Editors assigned to your paper RSOS-191511 "Foraging niche overlap during chick-rearing in the sexually dimorphic Westland petrel" have now received comments from reviewers and would like you to revise the paper in accordance with the reviewer comments and any comments from the Editors. Please note this decision does not guarantee eventual acceptance.

Please submit your revised manuscript and required files (see below) no later than 21 days from today's (ie 10-Sep-2020) date. Note: the ScholarOne system will 'lock' if submission of the revision is attempted 21 or more days after the deadline. If you do not think you will be able to meet this deadline please contact the editorial office immediately.

on behalf of Dr Agustina Gómez-Laich (Associate Editor) and Pete Smith (Subject Editor)

Associate Editor Comments to Author (Dr Agustina Gómez-Laich):

I found the work interesting and I commend the authors for their efforts and for presenting information on a procellariiform species with such an unusually strategy. The data presented broadens the general understanding of the species foraging ecology and provides information that could be employed for its conservation. Both reviewers found the paper well written and interesting, however, both have suggestions principally for the introduction and discussion. Adding some hypothesis and/or predictions would improve the structure of the manuscript and would help to broaden the scope. The introduction could be substantially improved by instead of focusing on procellariiforms common characteristics, focus on procellariiform species that do not follow the general patterns. After this authors could present possible explanations of the evolution of these alternative behaviors and present the Westland petrel as a case of study. These changes in the introduction would benefit the discussion. Instead of making so many comparisons authors could concentrate on possible explanations to the patterns they observed. It would also be interesting to see a more integrated discussion of the topics and avoid repeating information that was presented in the results.

I also have some specific comments that are listed below:

Materials and methods.

Page 4, last line. To “remove” erroneous locations? A word seems to be missing here.

Page 5, line 6. Why do authors use the maximum convex polygon to estimate the home range and not kernel density estimators?

Page 5, line 22. Please indicate in which program or environment this package was ran.

Page 5, line 42. Dynamic of each channel.

Page 5, line 47. Please indicate all the behaviors for which VEDBA was calculated (flapping, non flapping, rafting and diving?)

Page 5, line 48. Please mention the influential oceanographic variables.

Page 5, line 51. Please define foraging locations. Are these locations classified as dives? Dives plus at sea surface periods?

Page 5, line 56. Please indicate if this is an R package.

Page 6, line 15. How did you calculate the foraging rate?

Page 6, line 16. Did authors evaluate normality and homoscedasticity before applying an Anova test? Sex and year were tested as additive effects? Did authors test if the interaction was significant? How were significant variables detected in these Anova tests? Please incorporate this information.

Results.

Page 6, Line 59-60. Authors say the data presented in this study was combined with data from a previous study. Could you please give more information of how the data from both studies was combined, from which years Waught et al. 2018 data was, from how many animals. All this information should be mentioned in the methods section.

Figure 2. What do each green color represent? Different years? If each green color indicates a year why is there only one blue color if data comes from two years (2012 and 2015)? I find a bit difficult to understand this figure.

Table 1. I did not find a reference for table 1 in the text.

Table 2. Please define the Storer’s dimorphism index that is mentioned in Table 2 in the methodology section?

Page 7, line 1-4. The total distance traveled did not differ between years but the maximum distance did. It would be interesting to present a possible explanation to this.

Page 7, line 54-55. Authors stated that feeding conditions revealed by isotope analyses related to years of medium and high fledgling success. Do they have fledgling success from other years to make the comparisons? What does the number between brackets mean? Is it the number of nests checked?

Discussion

Page 7, lines 44-48. Does this mean that the same individual on consecutive trips gained and lost weight or does the data come from different individuals? The observed pattern could be associated to some foraging trips being more successful in terms of food acquisition than others.
 Page 9, line 22. Higher than “that” reported. I believe that “that” is missing here.
 Page 9, line 28. This is not in agreement with what is mentioned below. Based on the isotope analyses Westland petrels would be principally consuming fish.

Reviewer comments to Author:

Reviewer: 1

Comments to the Author(s)

Globally the manuscript is well written, with a good English, which allows a fluid and comprehensive reading. All sections have enough detail to be fully understood.

Major comments:

At the end of the introduction, after describing the main objectives of their study, the authors should raise few study hypothesis followed by the corresponding expected results. It's true the existing knowledge on the foraging ecology of their study species is scarce, but there is quite a few number of studies focusing on other (similar) gadfly petrels in other study areas. Then, latter in the results and discussion, the reader will be able to check whether the results confirmed the expectations, or the authors advance and discuss an alternative reasoning behind the foraging ecology choices of Westland petrels.

P5L48-56: I realise the authors describe in detail at the supplementary Table S1 the origin of the oceanographic variables used on their work, though they should also briefly describe here the spatio-temporal resolution of those environmental predictors.

Minor corrections:

P2L54: Please replace by “Procellariiforms”
 Change accordingly elsewhere in the main body of the manuscript.

P3L2: “Procellariiforms”

P3L27: “...during the chick-rearing period...”

P3L33: Replace “mediate” by “buffer”

P3L46: “...different threats, like seabird by-catch (Gianuca...”

P3L47: Remove “A” at the beginning of the sentence

P5L2: Replace “The distribution of the trip duration distribution was assessed for bimodality...” by “Trip duration distribution was assessed for bimodality...”

P6L26: Rewrite as “...by each sex.”

P7L45 and L46: remove "in area", it's kind of redundant

P7L48: "...current strength."

P10L10: Add a space before "Hence..."

P10L40: "...off the West coast of Africa..."

P10L60: "...strong El Niño/ La Niña..."

P17 Table 2: Please add an extra column with the t-test values before the column with P-values.

P24 Figure 4: Add to the legend what the two lines and CI are representing. Males and females? Change the colours; these are too similar to be easily distinguishable.

Reviewer: 2

Comments to the Author(s)

RSOS-191511

"Foraging niche overlap during chick-rearing in the sexually dimorphic Westland petrel"
by Poupart T.A. et al.

This study examined sexual differences in foraging behavior of Westland petrels in details. Somewhat, the scope is quite narrow, likely a description of specific behavior in a single species and not solely contributed to understanding of the general ecology of Procellariiform seabirds. However, otherwise, the manuscript is overall well and carefully written referring to previous knowledge efficiently, and methods and analysis are comprehensive and adequate. Furthermore, discussion on the lack of sexual differences in behavior in terms of habitat quality is relatively speculative based on Louzao et al. (2011), but still understandable. So, I highly appreciate this study and believe that this manuscript can be suitable for publication in Royal Society Open Science, but it requires some reconsideration.

Specific comment

INTRODUCTION

P2 "A few Procellariiform species (n=12) differ...throughout the winter."

>The general ecology of seabirds/Procellariiforms is summarized before this sentence. These descriptions are true, but only related to a part of this study, such as size dimorphism and dual foraging strategy. For example, results of this study do not provide an insight into why Westland petrels breed in winter. So, I suggest to shorten these sentences. Especially, although I like it, the first sentence describing the diversity of seabirds does not seem necessary (too general and not related to results of this study).

DISCUSSION

P7 "without...increased foraging effort compared to summer breeding species."

>Body size, prey species, and foraging range etc. are different to species mentioned (Scopoli's shearwater and Buller's albatross). So, foraging effort cannot be simply comparable to that of Westland petrels.

P7 "both positive and negative mass changes...self-maintenance in the same area."

>Were there any differences in foraging behavior between trips with positive and negative mass changes?

P7 "Information on the activity...their energy expenditure,"

>I wonder why the authors examined VeDBA in this study. Probably, you can find results in previous studies, showing differences in energy expenditure among behaviors in congeneric or similar species (foraging, flapping, gliding, rafting). I would like to see some discussion for VeDBA, recorded for Westland petrels in concert with GPS data. Also, did overall VeDBA differ between years with different sea surface temperature conditions? Such information could be useful to predict the effect of environmental changes on their breeding success.

===PREPARING YOUR MANUSCRIPT===

- one version identifying all the changes that have been made (for instance, in coloured highlight, in bold text, or tracked changes);
- a 'clean' version of the new manuscript that incorporates the changes made, but does not highlight them. This version will be used for typesetting if your manuscript is accepted.

===PREPARING YOUR REVISION IN SCHOLARONE===

- 1) One version identifying all the changes that have been made (for instance, in coloured highlight, in bold text, or tracked changes);
 - 2) A 'clean' version of the new manuscript that incorporates the changes made, but does not highlight them.
 - An individual file of each figure (EPS or print-quality PDF preferred [either format should be produced directly from original creation package], or original software format).
 - An editable file of each table (.doc, .docx, .xls, .xlsx, or .csv).
 - An editable file of all figure and table captions.
- Note: you may upload the figure, table, and caption files in a single Zip folder.
- Any electronic supplementary material (ESM).
 - If you are requesting a discretionary waiver for the article processing charge, the waiver form must be included at this step.
 - If you are providing image files for potential cover images, please upload these at this step, and inform the editorial office you have done so. You must hold the copyright to any image provided.
 - A copy of your point-by-point response to referees and Editors. This will expedite the preparation of your proof.

- Ensure that your data access statement meets the requirements at <https://royalsociety.org/journals/authors/author-guidelines/#data>. You should ensure that you cite the dataset in your reference list. If you have deposited data etc in the Dryad repository, please include both the 'For publication' link and 'For review' link at this stage.
- If you are requesting an article processing charge waiver, you must select the relevant waiver option (if requesting a discretionary waiver, the form should have been uploaded at Step 3 'File upload' above).
- If you have uploaded ESM files, please ensure you follow the guidance at <https://royalsociety.org/journals/authors/author-guidelines/#supplementary-material> to include a suitable title and informative caption. An example of appropriate titling and captioning may be found at https://figshare.com/articles/Table_S2_from_Is_there_a_trade-off_between_peak_performance_and_performance_breadth_across_temperatures_for_aerobic_sc_ope_in_teleost_fishes_/3843624.

Author's Response to Decision Letter for (RSOS-191511.R0)

See Appendix A.

Decision letter (RSOS-191511.R1)

Dear Mr Poupart

On behalf of the Editors, we are pleased to inform you that your Manuscript RSOS-191511.R1 "Foraging niche overlap during chick-rearing in the sexually dimorphic Westland petrel" has been accepted for publication in Royal Society Open Science subject to minor revision in accordance with the referees' reports. Please find the referees' comments along with any feedback from the Editors below my signature.

Please submit your revised manuscript and required files (see below) no later than 7 days from today's (ie 15-Oct-2020) date. Note: the ScholarOne system will 'lock' if submission of the revision is attempted 7 or more days after the deadline. If you do not think you will be able to meet this deadline please contact the editorial office immediately.

on behalf of Dr Agustina Gómez-Laich (Associate Editor) and Pete Smith (Subject Editor)
openscience@royalsociety.org

Associate Editor Comments to Author (Dr Agustina Gómez-Laich):
Comments to the Author:

I have only a few minor comments and suggestions that are listed below. Specific comments relate to the page and line number of the clean version of the word document that was available for review.

Abstract.

Pag 1, line 17. Please add the word behaviour after foraging.

Pag 1, line 24-25. I suggest replacing "without winter increase of the foraging effort or intra-specific competition avoidance" by "without increasing the foraging effort or intra-specific competition avoidance during winter".

Introduction.

Pag 1, line 34. There is an extra period here.

Pag 1, line 35. There is an extra parenthesis before "Prince"

Pag 2, line 26. A space is missing before "Another".

Pag 2, line 49. Instead of "they have", I suggest "this species has"

Pag 2, line 49. Instead of “shorter trips” I suggest “and performs shorter trips”

Pag 3, line 5. I suggest replacing “Fine scale foraging for this species” by “Fine scale foraging behaviour of this species”.

Pag 3, line 6. There is an extra space after its.

Pag 3, line 6. “obtained” instead of “obtain”

Materials and methods.

Pag 4, line 10. I suggest “Each” nest instead of “The” nests.

Pag 5, line 18. Why is this Supplementary Fig 3 if it is mentioned before Supplementary Fig 2?

Pag 5, line 28. Here authors mentioned they identified the following behaviours: flapping flight, soaring flight, rafting on water, foraging. However afterwards they say foraging included both diving and foraging on water. Having this in mind, when they first mentioned the recognized behaviours foraging should be divided into two: diving and foraging on water

Pag 5, line 39. In “an” area?

Discussion

Pag 8, line 17. Please remove the period or the comma.

Pag 8, line 24. Than “the” Westland petrel.

Pag 9, line 2. “is a” instead of “isa”.

Pag 9, line 3. There is an extra space before albatrosses.

Pag 9, line 4. There is an extra period after the parenthesis.

Pag 9, line 25-26. I suggest combining these two sentences. “The peak observed during the first hours of darkness suggests a pulsed food availability at ”

Pag 9, line 34-35. Prey is mentioned two times in this sentence.

Pag 9, line 45. “, in the present study it increased” should be deleted I believe.

Pag 10, line 30. Please correct the comma inside the reference parenthesis.

Pag 11, line 24. Proposed for “the” Westland petrel.

Pag 11, line 25. Fishing area “coincides”

Pag 11, line 39. Singular “foraging ecology”.

Figure 2 legend. This figure legend is not clear. Please modify it.

Figure 5. I suggest using the same colors as in figure 3 and 4. That is, males in blue and females in orange.

Figure 1. “Flying” and “rafting on water” of pannel A are written in a smaller size than “Flying”, “Foraging dive”, “Rafting on water”, “Flying” of pannel C. Also the font in which the time is written seems to be different between pannels.

===PREPARING YOUR MANUSCRIPT===

- one version identifying all the changes that have been made (for instance, in coloured highlight, in bold text, or tracked changes);
- a 'clean' version of the new manuscript that incorporates the changes made, but does not highlight them.

This version will be used for typesetting.

===PREPARING YOUR REVISION IN SCHOLARONE===

Author's Response to Decision Letter for (RSOS-191511.R1)

See Appendix B.

Decision letter (RSOS-191511.R2)

Dear Mr Poupart,

It is a pleasure to accept your manuscript entitled "Foraging niche overlap during chick-rearing in the sexually dimorphic Westland petrel" in its current form for publication in Royal Society Open Science. The comments of the reviewer(s) who reviewed your manuscript are included at the foot of this letter.

on behalf of Dr Agustina Gómez-Laich (Associate Editor) and Pete Smith (Subject Editor)
openscience@royalsociety.org

Associate Editor Comments to Author (Dr Agustina Gómez-Laich):
Associate Editor
Comments to the Author:
I only have one additional comment.
In Page 16, line 19 the word "behaviour" should not be deleted.

Appendix A

Associate Editor Comments to Author (Dr Agustina Gómez-Laich)		
Section / location in the text	Comment	Answer
Introduction	Adding some hypothesis and/or predictions would improve the structure of the manuscript and would help to broaden the scope. The introduction could be substantially improved by instead of focusing on procellariiforms common characteristics, focus on procellariiform species that do not follow the general patterns . After this, authors could present possible explanations of the evolution of these alternative behaviors and present the Westland petrel as a case of study	The introduction, pointed by all reviewers, has been partially reorganised and changed to take in account the comments. Now the common characteristics of Procellariiforms is shorter, to focus more on the winter-breeding ones, their paradoxes, the interest of their study, and finally the hypothesis / expected results on the Westland petrel as a study case.
Discussion	Instead of making so many comparisons authors could concentrate on possible explanations to the patterns they observed . It would also be interesting to see a more integrated discussion of the topics and avoid repeating information that was presented in the results.	The discussion has been shortened a little, amended, and the last paragraph has been augmented in order to give more possible explanation with a broader scope than the study species itself.
Page 4, last line.	To “remove” erroneous locations? A word seems to be missing here.	Yes, “remove” added
Page 5, line 6.	Why do authors use the maximum convex polygon to estimate the	Contrary to the kernels, the MCP do not suffer of the influence of the smoothing factor on the estimated area for comparisons (Row &

	home range and not kernel density estimators?	Blouin-Demers, 2006). That's why it was preferred. Reference: Jeffrey R. Row, & Blouin-Demers, G. (2006). Kernels Are Not Accurate Estimators of Home-Range Size for Herpetofauna. Copeia , 2006(4), 797-802.
Page 5, line 22.	Please indicate in which program or environment this package was ran.	Igor Pro software indication added
Page 5, line 42.	Dynamic of each channel.	"in each channel" added
Page 5, line 47.	Please indicate all the behaviors for which VEDBA was calculated (flapping, non flapping, rafting and diving?)	Indication of behaviours added: "flapping flight, soaring flight, rafting on water, foraging"
Page 5, line 48	Please mention the influential oceanographic variables.	Variables added: "Static variables such as depth and seafloor slope, and dynamic variables of the sea surface such as chlorophyll a concentration, temperature and its anomaly, height, water velocity, wave height and wind were obtained at spatial resolution between 0.01° - 0.25° and temporal resolution between an hour and a day"
Page 5, line 51.	Please define foraging locations. Are these locations classified as dives? Dives plus at sea surface periods?	Foraging definition added : "(regrouping foraging identified on water and diving)"
Page 5, line 56.	Please indicate if this is an R package.	Yes it is, indication added: "using the R package raster"
Page 6, line 15.	How did you calculate the foraging rate?	Definition of the foraging rate added: "(number of foraging events divided by the trip duration)"
Page 6, line 16.	Did authors evaluate normality and homoscedasticity before applying an Anova test? Sex and year were tested as additive effects? Did authors test if the interaction was significant? How were significant variables detected in these Anova tests? Please incorporate this information.	The validity of the ANOVA models was checked with the examination of the residuals (Zuur et al. 2009). This information has been added in the methods section. Sex and year were only tested only as additive effects. Given the small sample size of sexes within years, the interaction was not tested to avoid presenting spurious results. ANOVA results are already presented with the associated P value
Page 6, Line 59-60.	Authors say the data presented in this study	Information added at the beginning of the data processing and statistical analysis section: "

	was combined with data from a previous study. Could you please give more information of how the data from both studies was combined, from which years Waught et al. 2018 data was, from how many animals. All this information should be mentioned in the methods section.	Additional trip duration data were obtained during chick-rearing period, from respectively 29 trips in 2012 and 6 trips in 2015 (Waugh et al. 2018).”
Figure 2.	What do each green color represent? Different years? If each green color indicates a year why is there only one blue color if data comes from two years (2012 and 2015)? I find a bit difficult to understand this figure.	That is right, the different colors were hard to differentiate, and the distinction between 2012 et 2015 was missing. Hence, I changed this figure for cumulated bars, with one color (white, light grey, dark grey and black) by years so it easier to understand the figure.
Table 1.	I did not find a reference for table 1 in the text.	Reference to the Table 1 is present at the beginning of the results section: “ data were obtained from 29 individuals (16 males / 18 trips, 13 females / 15 trips, Error! Reference source not found.) totalling 1,556 h in duration”
Table 2.	Please define the Storer’s dimorphism index that is mentioned in Table 2 in the methodology section?	Description of this index added at the beginning of the methods section:” The degree of sexual size dimorphism (SSD) was quantified for each morphometric parameters using the Storer’s index: $SSD = 100 \times \frac{\text{mean } \sigma - \text{mean } \varphi}{(\text{mean } \sigma + \text{mean } \varphi) \times 0.5}$ (Storer 1966). This percentage of the relative difference in parameter between sexes has the advantage to avoid a size effect in the measure of SSD (Benito and González- Solís, 2007).”
Page 7, line 1-4.	The total distance traveled did not differ between years but the maximum distance did. It would be interesting to present a possible explanation to this.	Explanation by the difference in wind conditions and the seabirds movements adaptation to it : “ The difference in wind speed (5.5 ± 0.08 m.s-1 in 2016, 8.1 ± 0.08 m.s-1 in 2017) could explain the further maximum distance, higher trip speed and more time spent flying in 2017. Hence, the absence of difference in total horizontal distance parcoured between years seems to indicate an easier prey accessibility in 2017,

		and suggests some behavioural plasticity to compensate for variation of the energetic costs of movements for foraging (Wilson et al. 2012, Elliott et al. 2014, Ventura et al. 2020).”
Page 7, line 54-55	Authors stated that feeding conditions revealed by isotope analyses related to years of medium and high fledgling success. Do they have fledgling success from other years to make the comparisons? What does the number between brackets mean? Is it the number of nests checked?	Yes, the number between brackets showed the number of monitored nests at the colony. Since they can confuse the reader, I replaced them by the reference to a publication about nest monitoring. Yes the fledgling success is available for other years, I added this information as follow: “ These feeding conditions related to low fledgling success in 2016 (51 %) and high fledgling success in 2017 (85 %), as indicated by the burrows monitoring at the colony with an average success of 62 % (Waugh et al. 2006).”
Page 7, lines 44-48.	Does this mean that the same individual on consecutive trips gained and lost weight or does the data come from different individuals? The observed pattern could be associated to some foraging trips being more successful in terms of food acquisition than others.	The few individuals which made two consecutive trips (n=5) did it because we missed their return at the nest. As we have only the weight before and after the trip(s), in case of consecutive trips, the mass change is confounded within the two trips, without the possibility to know which one was more or less profitable (and foraging events by the accelerometer do not inform on the prey size). Hence, the data come from different individuals.
Page 9, line 22.	Higher than “that” reported. I believe that “that” is missing here.	“that” added
Page 9, line 28.	This is not in agreement with what is mentioned below. Based on the isotope analyses Westland petrels would be principally consuming fish.	If Westland petrel consume principally fish, Crustacea such as Nyctiphanes australis is also found in its diet (Freeman 1998). The goal of this sentence is to show that our findings on foraging is in accordance with the distribution and ecology of its prey species. So this sentence was amended as follow: “ The findings suggest Westland petrels feed on prey patchily and aggregated prey distributed throughout the shelf slope, in accordance with what is known about the prey species of its diet (O'brien 1988, Freeman 1998).” Reference: Freeman, A. N. (1998). Diet of Westland Petrels Procellaria westlandica : The importance of fisheries waste during chick-rearing. Emu , 98(1), 36-43.

Reviewer 1		
Introduction	At the end of the introduction, after describing the main objectives of their study, the authors should raise few study hypothesis followed by the corresponding expected results.	The introduction, pointed by all reviewers, has been partially reorganised and changed to take in account the comments. Now the common characteristics of Procellariiforms is shorter, to focus more on the winter-breeding ones, their paradoxes, the interest of their study, and finally the hypothesis / expected results on the Westland petrel as a study case.
Results and discussion	results and discussion, the reader will be able to check whether the results confirmed the expectations	The hypothesis and expected results has been exposed more clearly in the introduction, so the following results and discussion are clearer now to show whether or not the present results confirmed the expectations.
P5 L48-56	I realise the authors describe in detail at the supplementary Table S1 the origin of the oceanographic variables used on their work, though they should also briefly describe here the spatio-temporal resolution of those environmental predictors.	In addition to the Table S1, the spatio-temporal resolution was added briefly in the text: "Static variables such as depth and seafloor slope, and dynamic variables of the sea surface such as chlorophyll a concentration, temperature and its anomaly, height, water velocity, wave height and wind were obtained at spatial resolution between 0.01° - 0.25° and temporal resolution between an hour and a day"
P2 L54	Please replace by "Procellariiforms" Change accordingly elsewhere in the main body of the manuscript.	Typo corrected, "i" added on all mistaken words detected with an automatic research in the text.
P3 L2	"Procellariiforms"	Typo corrected
P3 L27	"...during the chick-rearing period..."	"the" added
P3 L33	Replace "mediate" by "buffer"	Done
P3 L46	"...different threats, like seabird by-catch (Gianuca..."	Precision "like seabird by-catch" added
P3 L47	Remove "A" at the beginning of the sentence	"A" removed
P5 L2	Replace "The distribution of the trip duration distribution was assessed for bimodality..." by "Trip duration distribution was	done

	assessed for bimodality...”	
P6 L26	Rewrite as “...by each sex.”	done
P7 L45 and L46:	remove “in area”, it’s kind of redundant	Done
P7 L48	“...current strength.”	Changed for “speed”, for consistency with the environmental data table.
P10 L10	Add a space before “Hence...”	Typo corrected
P10 L40	“...off the West coast of Africa...”	Typo corrected
P10 L60	“...strong El Niño/ La Niña...”	Typo corrected
P17 Table 2	Please add an extra column with the t-test values before the column with P-values.	T-test values added in an extra column
P24 Figure 4	Add to the legend what the two lines and CI are representing. Males and females? Change the colours; these are too similar to be easily distinguishable.	Figure 4 has been updated with more distinguishable colors, also adapted to colorblindness. The legend has been amended to explain the lines and the confidence interval.
Reviewer 2		
Introduction	”A few Procellariform speces (n=12) differ...throughout the winter.” >The general ecology of seabirds/Procellariforms is summarized before this sentence. These descriptions are true, but only related to a part of this study, such as size dimorphism and dual foraging strategy. For example, results of this study do not provide an insight into why Westland petrels breed in winter. So, I suggest to shorten these sentences. Especially, although I like it, the first sentence describing the diversity of seabirds does not seem necessary (too	The introduction, pointed by all reviewers, has been partially reorganised and changed to take in account the comments. Now the common characteristics of Procellariiforms is shorter, to focus more on the winter-breeding ones, their paradoxe, the interest of their study, and finally the hypothesis / expected results on the Westland petrel as a study case.

	general and not related to results of this study).																													
DISCUSSION P7	“without...increased foraging effort compared to summer breeding species.” >Body size, prey species, and foraging range etc. are different to species mentioned (Scopoli’s shearwater and Buller’s albatross). So, foraging effort cannot be simply comparable to that of Westland petrels.	Despite accelerometry is a growing research topic, acceleration data at the scale of the foraging trips is still lacking for most seabird species. Hence, I refer to what is available in Procellariiforms. Off course species have different size, prey, foraging techniques and field metabolic rates. But showing their respective mean VeDBA or foraging rate, in complement of other trips parameters (trip length, duration) allows some comparative approach in order to identify proxy of higher / lower foraging effort.																												
DISCUSSION P7	“both positive and negative mass changes...self-maintenance in the same area.” >Were there any differences in foraging behavior between trips with positive and negative mass changes?	No, there was no significant change between in the track parameters between positive and negative mass changes. They are presented below :     Positive Negative ANOVA P value     Max. distance to the colony (km) 166 ± 19 150 ± 26 0.6   Mean speed (km/h) 19 ± 1.4 16 ± 1.5 0.2   Total distance (km) 1040 ± 121 885 ± 179 0.5   Duration (h) 56 ± 6 50 ± 8 0.6   Mean VeDBA (g) 0.222 ± 0.007 0.228 ± 0.006 0.5   Foraging rate (events/h) 3.1 ± 0.5 2.5 ± 0.4 0.3    During fieldwork, the burrows of the equipped birds were checked every 2 hours through the night for recapture and logger retrieval. Hence, it is very likely that some of them were recaptured after feeding their chick, biasing the mass change associated to the at-sea trip. Therefore, we choose to stay careful with these results, not showing them and recommending further study with more		Positive	Negative	ANOVA P value	Max. distance to the colony (km)	166 ± 19	150 ± 26	0.6	Mean speed (km/h)	19 ± 1.4	16 ± 1.5	0.2	Total distance (km)	1040 ± 121	885 ± 179	0.5	Duration (h)	56 ± 6	50 ± 8	0.6	Mean VeDBA (g)	0.222 ± 0.007	0.228 ± 0.006	0.5	Foraging rate (events/h)	3.1 ± 0.5	2.5 ± 0.4	0.3
	Positive	Negative	ANOVA P value																											
Max. distance to the colony (km)	166 ± 19	150 ± 26	0.6																											
Mean speed (km/h)	19 ± 1.4	16 ± 1.5	0.2																											
Total distance (km)	1040 ± 121	885 ± 179	0.5																											
Duration (h)	56 ± 6	50 ± 8	0.6																											
Mean VeDBA (g)	0.222 ± 0.007	0.228 ± 0.006	0.5																											
Foraging rate (events/h)	3.1 ± 0.5	2.5 ± 0.4	0.3																											

		accurate weighing: “ A more accurate weighing of the individuals, just after their return at the colony, is required to verify this potential explanation for such contrasted mass changes.”
DISCUSSION P7	“Information on the activity...their energy expenditure,” >I wonder why the authors examined VeDBA in this study. Probably, you can find results in previous studies, showing differences in energy expenditure among behaviors in congeneric or similar species (foraging, flapping, gliding, rafting). I would like to see some discussion for VeDBA, recorded for Westland petrels in concert with GPS data. Also, did overall VeDBA differ between years with different sea surface temperature conditions? Such information could be useful to predict the effect of environmental changes on their breeding success.	In this study, the VeDBA was examined at different scale to bring three kind of information. At the second scale, VeDBA was used 1) to find accurately in space and time the foraging behaviour during the trips. In addition, this proxy of the field metabolic rate was used 2) to compare the cost of all the behaviours recognised (flapping, gliding, rafting and foraging) as this information was not available for the study species. At the whole trip scale, the mean VeDBA was used as proxy of foraging effort, 3) to inform on how hard it is to forage in winter. Hence, VeDBA results were more explicitly presented, with addition of the mean VeDBA values in the results section, table 1, and model in table S2. The foraging trip mean VeDBA didn't significantly differed between the years of study, and the mean VeDBA is further discussed in the discussion : “ This foraging behaviour was found constant in all trips for both sexes, as shown by the mean VeDBA and foraging rate at the trip scale. This is in contrast with the distinction between fine-scale and coarse-scale foraging trips reported in Scopoli's shearwater Calonectris diomedea”
Figure 3		For consistency with the figure 4, where more distinct colours were requested by a reviewer, the same change has been made on the figure 3.
Bibliography		The bibliography has been updated according to the changes, and formatted to fit the standards of the journal.

Appendix B

Abstract		
Pag 1, line 17	Please add the word behaviour after foraging.	Done
Pag 1, line 24-25	I suggest replacing “without winter increase of the foraging effort or intra-specific competition avoidance” by “without increasing the foraging effort or intra-specific competition avoidance during winter”.	Done
Introduction		
Pag 1, line 34	There is an extra period here	Removed
Pag 1, line 35	There is an extra parenthesis before “Prince”	Removed
Page 2, line 26	A space is missing before “Another”	Added
Pag 2, line 49	Instead of “they have”, I suggest “this species has”	Changed
Pag 2, line 49	Instead of “shorter trips” I suggest “and performs shorter trips”	“performs” added
Pag 3, line 5	I suggest replacing “Fine scale foraging for this species” by “Fine scale foraging behaviour of this species”.	changed
Pag 3, line 6	There is an extra space after its	removed
Pag 3, line 6	“obtained” instead of “obtain”	Changed
Materials and methods		
Pag 4, line 10	I suggest “Each” nest instead of “The” nests	changed
Pag 5, line 18	Why is this Supplementary Fig 3 if it is mentioned before Supplementary Fig 2?	True, order changed to respect logical ascending order
Pag 5, line 28	Here authors mentioned they identified the following behaviours: flapping flight, soaring flight, rafting on water, foraging. However afterwards they say foraging included both diving and foraging on water. Having this in mind, when they first mentioned the recognized behaviours foraging should be divided into two: diving and foraging on water	The first mention of the recognised behaviour has been amended as follow to state more clearly the inclusion of on-water and diving in foraging behaviour: “At-sea behaviours (flapping flight, soaring flight, rafting on water, foraging on water and diving) were inferred at each second of the trip using the acceleration data, in order to access the individual’s fine-scale activity budgets and spatio-temporal localisation of foraging behaviour.”

Pag 5, line 39	In “an” area?	Apparently the dedicated expression is “time in area” (Warwick Evans et al. 2015) So I left it like it was. Reference: Warwick-Evans, V., Atkinson, P. W., Gauvain, R. D., Robinson, L. A., Arnould, J. P. Y., & Green, J. A. (2015). Time-in-area represents foraging activity in a wide-ranging pelagic forager. Mar Ecol Prog Ser , 527, 233-246.
Discussion		
Pag 8, line 17	Please remove the period or the comma	Comma removed
Pag 8, line 24	Than “the” Westland petrel.	The added
Pag 9, line 2	“is a” instead of “isa”	Space added
Pag 9, line 3	There is an extra space before albatrosses	Removed
Pag 9, line 4	The is an extra period after the parenthesis	Removed
Pag 9, line 25-26	I suggest combining these two sentences. “The peak observed during the first hours of darkness suggests a pulsed food availability at “	Sentence corrected as suggested
Pag 9, line 34-35	Prey is mentioned two times in this sentence.	One was removed
Pag 9, line 45	“, in the present study it increased” should be deleted I believe.	Yes, that is right.
Pag 10, line 30	Please correct the comma inside the reference parenthesis.	Comma removed
Pag 11, line 24	Proposed for “the” Westland petrel.	added
Pag 11, line 25	Fishing area “coincides”	Changed as proposed
Pag 11, line 39	Singular “foraging ecology”	Changed as proposed
Figure 2 legend.	This figure legend is not clear. Please modify it.	True, the figure was changed at the previous revision, but not the caption. Hence, the legend has been modified as follow: “ Distribution of the foraging trip durations of chick-rearing Westland petrels. Data from 2016 and 2017 were gathered during the present study, data from ”

		2012 and 2015 are from Waugh et al. (2018)."
Figure 5.	I suggest using the same colors as in figure 3 and 4. That is, males in blue and females in orange.	Colors changed as proposed for consistency with figure 3 and 4
S Figure 1.	"Flying" and "rafting on water" of pannel A are writen in a smaller size than "Flying", "Foraging dive", "Rafting on water", "Flying" of pannel C. Also the font in which the time is written seems to be different between pannels.	Text size has been homogeneized between panels. Pannels A, B and C show different time scales, that is why the hours are labelled on each panel, while the general and common time legend is written only once below (time of the day).
Bibliography		
		Missing references has been added (Forest & Bird 2014). Journals title updated with abbreviations and italic format